# Deep Complex Networks

**Chiheb Trabelsi,**[*♦♣] **Olexa Bilaniuk,**[*♦] **Ying Zhang,**[†♦♠] **Dmitriy Serdyuk,**[†♦] **Sandeep Subramanian,**[†♦]

**João Felipe Santos,**[♦] **Soroush Mehri,**[♥] **Negar Rostamzadeh,**[♠] **Yoshua Bengio**[♦¶] **& Christopher J Pal**[♦♣]

[♦] Montreal Institute for Learning Algorithms (MILA), Montreal
[♣] Ecole Polytechnique, Montreal
[♥] Microsoft Research, Montreal
[♠] Element AI, Montreal
[¶] CIFAR Senior Fellow
`chiheb.trabelsi@polymtl.ca, olexa.bilaniuk@umontreal.ca, ying.zhlisa@gmail.com,`
`{serdyuk@iro, sandeep.subramanian.1@}umontreal.ca, jfsantos@emt.inrs.ca,`
`soroush.mehri@microsoft.com, negar@elementai.com,`
`find.me@the.web, christopher.pal@polymtl.ca`

## Abstract

At present, the vast majority of building blocks, techniques, and architectures for deep learning are based on real-valued operations and representations. However, recent work on recurrent neural networks and older fundamental theoretical analysis suggests that complex numbers could have a richer representational capacity and could also facilitate noise-robust memory retrieval mechanisms. Despite their attractive properties and potential for opening up entirely new neural architectures, complex-valued deep neural networks have been marginalized due to the absence of the building blocks required to design such models. In this work, we provide the key atomic components for complex-valued deep neural networks and apply them to convolutional feed-forward networks and convolutional LSTMs. More precisely, we rely on complex convolutions and present algorithms for complex batch-normalization, complex weight initialization strategies for complex-valued neural nets and we use them in experiments with end-to-end training schemes. We demonstrate that such complex-valued models are competitive with their real-valued counterparts. We test deep complex models on several computer vision tasks, on music transcription using the MusicNet dataset and on Speech Spectrum Prediction using the TIMIT dataset. We achieve state-of-the-art performance on these audio-related tasks.

## 1 Introduction

Recent research advances have made significant progress in addressing the difficulties involved in learning deep neural network architectures. Key innovations include normalization techniques (Ioffe and Szegedy, 2015; Salimans and Kingma, 2016) and the emergence of gating-based feed-forward neural networks like Highway Networks (Srivastava et al., 2015). Residual networks (He et al., 2015a; 2016) have emerged as one of the most popular and effective strategies for training very deep convolutional neural networks (CNNs). Both highway networks and residual networks facilitate the training of deep networks by providing shortcut paths for easy gradient flow to lower network layers thereby diminishing the effects of vanishing gradients (Hochreiter, 1991). He et al. (2016)

---

[*]Equal first author
[†]Equal contributions

show that learning explicit residuals of layers helps in avoiding the vanishing gradient problem and provides the network with an easier optimization problem. Batch normalization (Ioffe and Szegedy, 2015) demonstrates that standardizing the activations of intermediate layers in a network across a minibatch acts as a powerful regularizer as well as providing faster training and better convergence properties. Further, such techniques that standardize layer outputs become critical in deep architectures due to the vanishing and exploding gradient problems.

The role of representations based on complex numbers has started to receive increased attention, due to their potential to enable easier optimization (Nitta, 2002), better generalization characteristics (Hirose and Yoshida, 2012), faster learning (Arjovsky et al., 2015; Danihelka et al., 2016; Wisdom et al., 2016) and to allow for noise-robust memory mechanisms (Danihelka et al., 2016). Wisdom et al. (2016) and Arjovsky et al. (2015) show that using complex numbers in recurrent neural networks (RNNs) allows the network to have a richer representational capacity. Danihelka et al. (2016) present an LSTM (Hochreiter and Schmidhuber, 1997) architecture augmented with associative memory with complex-valued internal representations. Their work highlights the advantages of using complex-valued representations with respect to retrieval and insertion into an associative memory. In residual networks, the output of each block is added to the output history accumulated by summation until that point. An efficient retrieval mechanism could help to extract useful information and process it within the block.

In order to exploit the advantages offered by complex representations, we present a general formulation for the building components of complex-valued deep neural networks and apply it to the context of feed-forward convolutional networks and convolutional LSTMs. Our contributions in this paper are as follows:

1. A formulation of complex batch normalization, which is described in Section 3.5;

2. Complex weight initialization, which is presented in Section 3.6;

3. A comparison of different complex-valued ReLU-based activation functions presented in Section 4.1;

4. A state of the art result on the MusicNet multi-instrument music transcription dataset, presented in Section 4.2;

5. A state of the art result in the Speech Spectrum Prediction task on the TIMIT dataset, presented in Section 4.3.

We perform a sanity check of our deep complex network and demonstrate its effectiveness on standard image classification benchmarks, specifically, CIFAR-10, CIFAR-100. We also use a reduced-training set of SVHN that we call SVHN[*]. For audio-related tasks, we perform a music transcription task on the MusicNet dataset and a Speech Spectrum prediction task on TIMIT. The results obtained for vision classification tasks show that learning complex-valued representations results in performance that is competitive with the respective real-valued architectures. Our promising results in music transcription and speech spectrum prediction underscore the potential of deep complex-valued neural networks applied to acoustic related tasks[1] – We continue this paper with discussion of motivation for using complex operations and related work.

## 2 Motivation and Related Work

Using complex parameters has numerous advantages from computational, biological, and signal processing perspectives. From a computational point of view, Danihelka et al. (2016) has shown that Holographic Reduced Representations (Plate, 2003), which use complex numbers, are numerically efficient and stable in the context of information retrieval from an associative memory. Danihelka et al. (2016) insert key-value pairs in the associative memory by addition into a *memory trace*. Although not typically viewed as such, residual networks (He et al., 2015a; 2016) and Highway Networks (Srivastava et al., 2015) have a similar architecture to associative memories: each ResNet residual path computes a residual that is then inserted – by summing into the "memory" provided by the identity connection. Given residual networks' resounding success on several benchmarks and

---

[1]The source code is located at `http://github.com/ChihebTrabelsi/deep_complex_networks`

their functional similarity to associative memories, it seems interesting to marry both together. This motivates us to incorporate complex weights and activations in residual networks. Together, they offer a mechanism by which useful information may be retrieved, processed and inserted in each residual block.

Orthogonal weight matrices provide a novel angle of attack on the well-known vanishing and exploding gradient problems in RNNs. Unitary RNNs (Arjovsky et al., 2015) are based on unitary weight matrices, which are a complex generalization of orthogonal weight matrices. Compared to their orthogonal counterparts, unitary matrices provide a richer representation, for instance being capable of implementing the discrete Fourier transform, and thus of discovering spectral representations. Arjovsky et al. (2015) show the potential of this type of recurrent neural networks on toy tasks. Wisdom et al. (2016) provided a more general framework for learning unitary matrices and they applied their method on toy tasks and on a real-world speech task.

Using complex weights in neural networks also has biological motivation. Reichert and Serre (2013) have proposed a biologically plausible deep network that allows one to construct richer and more versatile representations using complex-valued neuronal units. The complex-valued formulation allows one to express the neuron's output in terms of its firing rate and the relative timing of its activity. The amplitude of the complex neuron represents the former and its phase the latter. Input neurons that have similar phases are called *synchronous* as they add constructively, whereas *asynchronous* neurons add destructively and thus interfere with each other. This is related to the gating mechanism used in both deep feed-forward neural networks (Srivastava et al., 2015; van den Oord et al., 2016a;b) and recurrent neural networks (Hochreiter and Schmidhuber, 1997; Cho et al., 2014; Zilly et al., 2016) as this mechanism learns to synchronize inputs that the network propagates at a given feed-forward layer or time step. In the context of deep gating-based networks, synchronization means the propagation of inputs whose controlling gates simultaneously hold high values. These controlling gates are usually the activations of a sigmoid function. This ability to take into account phase information might explain the effectiveness of incorporating complex-valued representations in the context of recurrent neural networks.

The phase component is not only important from a biological point of view but also from a signal processing perspective. It has been shown that the phase information in speech signals affects their intelligibility (Shi et al., 2006). Also Oppenheim and Lim (1981) show that the amount of information present in the phase of an image is sufficient to recover the majority of the information encoded in its magnitude. In fact, phase provides a detailed description of objects as it encodes shapes, edges, and orientations.

Recently, Rippel et al. (2015) leveraged the Fourier spectral representation for convolutional neural networks, providing a technique for parameterizing convolution kernel weights in the spectral domain, and performing pooling on the spectral representation of the signal. However, the authors avoid performing complex-valued convolutions, instead building from real-valued kernels in the spatial domain. In order to ensure that a complex parametrization in the spectral domain maps onto real-valued kernels, the authors impose a conjugate symmetry constraint on the spectral-domain weights, such that when the *inverse Fourier transform* is applied to them, it only yields real-valued kernels.

As pointed out in Reichert and Serre (2013), the use of complex-valued neural networks (Georgiou and Koutsougeras, 1992; Zemel et al., 1995; Kim and Adalı, 2003; Hirose, 2003; Nitta, 2004) has been investigated long before the earliest deep learning breakthroughs (Hinton et al., 2006; Bengio et al., 2007; Poultney et al., 2007). Recently Reichert and Serre (2013); Bruna et al. (2015); Arjovsky et al. (2015); Danihelka et al. (2016); Wisdom et al. (2016) have tried to bring more attention to the usefulness of deep complex neural networks by providing theoretical and mathematical motivation for using complex-valued deep networks. However, to the best of our knowledge, most of the recent works using complex valued networks have been applied on toy tasks, with the exception of some attempts. In fact, (Oyallon and Mallat, 2015; Tygert et al., 2015; Worrall et al., 2016) have used complex representation in vision tasks. Wisdom et al. (2016) have also performed a real-world speech task consisting of predicting the log magnitude of the future *short time Fourier transform* frames. In Natural Language Processing, (Trouillon et al., 2016; Trouillon and Nickel, 2017) have used complex-valued embeddings. Much remains to be done to develop proper tools and a general framework for training deep neural networks with complex-valued parameters.

Given the compelling reasons for using complex-valued representations, the absence of such frameworks represents a gap in machine learning tooling, which we fill by providing a set of building blocks for deep complex-valued neural networks that enable them to achieve competitive results with their real-valued counterparts on real-world tasks.

## 3 COMPLEX BUILDING BLOCKS

In this section, we present the core of our work, laying down the mathematical framework for implementing complex-valued building blocks of a deep neural network.

### 3.1 REPRESENTATION OF COMPLEX NUMBERS

We start by outlining the way in which complex numbers are represented in our framework. A complex number $z = a + ib$ has a real component $a$ and an imaginary component $b$. We represent the real part $a$ and the imaginary part $b$ of a complex number as logically distinct real valued entities and simulate complex arithmetic using real-valued arithmetic internally. Consider a typical real-valued $2D$ convolution layer that has $N$ feature maps such that $N$ is divisible by 2; to represent these as complex numbers, we allocate the *first* $N/2$ feature maps to represent the real components and the remaining $N/2$ to represent the imaginary ones. Thus, for a four dimensional weight tensor $W$ that links $N_{in}$ input feature maps to $N_{out}$ output feature maps and whose kernel size is $m \times m$ we would have a weight tensor of size $(N_{out} \times N_{in} \times m \times m)/2$ complex weights.

### 3.2 COMPLEX CONVOLUTION

In order to perform the equivalent of a traditional real-valued 2D convolution in the complex domain, we convolve a complex filter matrix $\mathbf{W} = \mathbf{A} + i\mathbf{B}$ by a complex vector $\mathbf{h} = \mathbf{x} + i\mathbf{y}$ where $\mathbf{A}$ and $\mathbf{B}$ are real matrices and $\mathbf{x}$ and $\mathbf{y}$ are real vectors since we are simulating complex arithmetic using real-valued entities. As the convolution operator is distributive, convolving the vector $\mathbf{h}$ by the filter $\mathbf{W}$ we obtain:

$$\mathbf{W} * \mathbf{h} = (\mathbf{A} * \mathbf{x} - \mathbf{B} * \mathbf{y}) + i(\mathbf{B} * \mathbf{x} + \mathbf{A} * \mathbf{y}). \tag{1}$$

As illustrated in Figure 1a, if we use matrix notation to represent real and imaginary parts of the convolution operation we have:

$$\begin{bmatrix} \Re(\mathbf{W} * \mathbf{h}) \\ \Im(\mathbf{W} * \mathbf{h}) \end{bmatrix} = \begin{bmatrix} \mathbf{A} & -\mathbf{B} \\ \mathbf{B} & \mathbf{A} \end{bmatrix} * \begin{bmatrix} \mathbf{x} \\ \mathbf{y} \end{bmatrix}. \tag{2}$$

### 3.3 COMPLEX DIFFERENTIABILITY

In order to perform backpropagation in a complex-valued neural network, a sufficient condition is to have a cost function and activations that are *differentiable* with respect to the real and imaginary parts of each complex parameter in the network. See Section 6.3 in the Appendix for the complex chain rule.

By constraining activation functions to be *complex differentiable* or *holomorphic*, we restrict the use of possible activation functions for a complex valued neural networks (For further details about holomorphism please refer to Section 6.2 in the appendix). Hirose and Yoshida (2012) shows that it is unnecessarily restrictive to limit oneself only to holomorphic activation functions; Those functions that are differentiable with respect to the real part and the imaginary part of each parameter are also compatible with backpropagation. (Arjovsky et al., 2015; Wisdom et al., 2016; Danihelka et al., 2016) have used non-holomorphic activation functions and optimized the network using regular, real-valued backpropagation to compute partial derivatives of the cost with respect to the real and imaginary parts.

Even though their use greatly restricts the set of potential activations, it is worth mentioning that holomorphic functions can be leveraged for computational efficiency purposes. As pointed out in Sarroff et al. (2015), using holomorphic functions allows one to share gradient values (because the activation satisfies the Cauchy-Riemann equations 11 and 12 in the appendix). So, instead of computing and backpropagating 4 different gradients, only 2 are required.

### 3.4 Complex-Valued Activations

#### 3.4.1 ModReLU

Numerous activation functions have been proposed in the literature in order to deal with complex-valued representations. (Arjovsky et al., 2015) have proposed modReLU, which is defined as follows:

$$\text{modReLU}(z) = \text{ReLU}(|z| + b)\, e^{i\theta_z} = \begin{cases} (|z| + b)\frac{z}{|z|} & \text{if } |z| + b \geq 0, \\ 0 & \text{otherwise,} \end{cases} \tag{3}$$

where $z \in \mathbb{C}$, $\theta_z$ is the phase of $z$, and $b \in \mathbb{R}$ is a learnable parameter. As $|z|$ is always positive, a bias $b$ is introduced in order to create a *"dead zone"* of radius $b$ around the origin $0$ where the neuron is inactive, and outside of which it is active. The authors have used modReLU in the context of unitary RNNs. Their design of modReLU is motivated by the fact that applying separate ReLUs on both real and imaginary parts of a neuron performs poorly on toy tasks. The intuition behind the design of modReLU is to preserve the pre-activated phase $\theta_z$, as altering it with an activation function severely impacts the complex-valued representation. modReLU does not satisfy the Cauchy-Riemann equations, and thus is not holomorphic. We have tested modReLU in deep feed-forward complex networks and the results are given in Table 6.4.

#### 3.4.2 ℂReLU and $z$ReLU

We call Complex ReLU (or ℂReLU) the complex activation that applies separate ReLUs on both of the real and the imaginary part of a neuron, i.e:

$$\mathbb{C}\text{ReLU}(z) = \text{ReLU}(\Re(z)) + i\,\text{ReLU}(\Im(z)). \tag{4}$$

ℂReLU satisfies the Cauchy-Riemann equations when both the real and imaginary parts are at the same time either strictly positive or strictly negative. This means that ℂReLU satisfies the Cauchy-Riemann equations when $\theta_z \in\, ]0,\, \pi/2[$ or $\theta_z \in\, ]\pi,\, 3\pi/2[$. We have tested ℂReLU in deep feed-forward neural networks and the results are given in Table 6.4.

It is also worthwhile to mention the work done by Guberman (2016) where a ReLU-based complex activation which satisfies the Cauchy-Riemann equations everywhere except for the set of points $\{\Re(z) > 0, \Im(z) = 0\} \cup \{\Re(z) = 0, \Im(z) > 0\}$ ias used. The activation function has similarities to ℂReLU. We call Guberman (2016) activation as $z$ReLU and is defined as follows:

$$z\text{ReLU}(z) = \begin{cases} z & \text{if } \theta_z \in [0, \pi/2], \\ 0 & \text{otherwise,} \end{cases} \tag{5}$$

We have tested $z$ReLU in deep feed-forward complex networks and the results are given in Table 6.4.

### 3.5 Complex Batch Normalization

Deep networks generally rely upon Batch Normalization (Ioffe and Szegedy, 2015) to accelerate learning. In some cases batch normalization is essential to optimize the model. The standard formulation of Batch Normalization applies only to real values. In this section, we propose a batch normalization formulation that can be applied for complex values.

To standardize an array of complex numbers to the standard normal complex distribution, it is not sufficient to translate and scale them such that their mean is 0 and their variance 1. This type of normalization does not ensure equal variance in both the real and imaginary components, and the resulting distribution is not guaranteed to be circular; It will be elliptical, potentially with high eccentricity.

We instead choose to treat this problem as one of whitening 2D vectors, which implies scaling the data by the square root of their variances along each of the two principal components. This can be done by multiplying the $\mathbf{0}$-centered data $(\boldsymbol{x} - \mathbb{E}[\boldsymbol{x}])$ by the inverse square root of the $2\times 2$ covariance matrix $\boldsymbol{V}$:

$$\tilde{\boldsymbol{x}} = (\boldsymbol{V})^{-\frac{1}{2}}\, (\boldsymbol{x} - \mathbb{E}[\boldsymbol{x}])\,,$$

where the covariance matrix $V$ is

$$V = \begin{pmatrix} V_{rr} & V_{ri} \\ V_{ir} & V_{ii} \end{pmatrix} = \begin{pmatrix} \text{Cov}(\Re\{x\}, \Re\{x\}) & \text{Cov}(\Re\{x\}, \Im\{x\}) \\ \text{Cov}(\Im\{x\}, \Re\{x\}) & \text{Cov}(\Im\{x\}, \Im\{x\}) \end{pmatrix}.$$

The square root and inverse of $2 \times 2$ matrices has an inexpensive, analytical solution, and its existence is guaranteed by the positive (semi-)definiteness of $V$. Positive definiteness of $V$ is ensured by the addition of $\epsilon I$ to $V$ (Tikhonov regularization). The mean subtraction and multiplication by the inverse square root of the variance ensures that $\tilde{x}$ has standard complex distribution with mean $\mu = 0$, covariance $\Gamma = 1$ and pseudo-covariance (also called relation) $C = 0$. The mean, the covariance and the pseudo-covariance are given by:

$$\begin{aligned} \mu &= \mathbb{E}\left[\tilde{x}\right] \\ \Gamma &= \mathbb{E}\left[(\tilde{x} - \mu)(\tilde{x} - \mu)^*\right] = V_{rr} + V_{ii} + i\left(V_{ir} - V_{ri}\right) \\ C &= \mathbb{E}\left[(\tilde{x} - \mu)(\tilde{x} - \mu)\right] = V_{rr} - V_{ii} + i\left(V_{ir} + V_{ri}\right). \end{aligned} \tag{6}$$

The normalization procedure allows one to decorrelate the imaginary and real parts of a unit. This has the advantage of avoiding co-adaptation between the two components which reduces the risk of overfitting (Cogswell et al., 2015; Srivastava et al., 2014).

Analogously to the real-valued batch normalization algorithm, we use two parameters, $\beta$ and $\gamma$. The shift parameter $\beta$ is a complex parameter with two learnable components (the real and imaginary means). The scaling parameter $\gamma$ is a $2 \times 2$ positive semi-definite matrix with only three degrees of freedom, and thus only three learnable components. In much the same way that the matrix $(V)^{-\frac{1}{2}}$ normalized the variance of the input to 1 along both of its original principal components, so does $\gamma$ scale the input along desired new principal components to achieve a desired variance. The scaling parameter $\gamma$ is given by:

$$\gamma = \begin{pmatrix} \gamma_{rr} & \gamma_{ri} \\ \gamma_{ri} & \gamma_{ii} \end{pmatrix}.$$

As the normalized input $\tilde{x}$ has real and imaginary variance 1, we initialize both $\gamma_{rr}$ and $\gamma_{ii}$ to $1/\sqrt{2}$ in order to obtain a modulus of 1 for the variance of the normalized value. $\gamma_{ri}$, $\Re\{\beta\}$ and $\Im\{\beta\}$ are initialized to 0. The complex batch normalization is defined as:

$$\text{BN}\left(\tilde{x}\right) = \gamma\,\tilde{x} + \beta. \tag{7}$$

We use running averages with momentum to maintain an estimate of the complex batch normalization statistics during training and testing. The moving averages of $V_{ri}$ and $\beta$ are initialized to 0. The moving averages of $V_{rr}$ and $V_{ii}$ are initialized to $1/\sqrt{2}$. The momentum for the moving averages is set to 0.9.

## 3.6 COMPLEX WEIGHT INITIALIZATION

In a general case, particularly when batch normalization is not performed, proper initialization is critical in reducing the risks of vanishing or exploding gradients. To do this, we follow the same steps as in Glorot and Bengio (2010) and He et al. (2015b) to derive the variance of the complex weight parameters.

A complex weight has a polar form as well as a rectangular form

$$W = |W|e^{i\theta} = \Re\{W\} + i\,\Im\{W\}, \tag{8}$$

where $\theta$ and $|W|$ are respectively the argument (phase) and magnitude of $W$.

Variance is the difference between the *expectation of the squared magnitude* and the *square of the expectation*:

$$\text{Var}(W) = \mathbb{E}\left[WW^*\right] - (\mathbb{E}\left[W\right])^2 = \mathbb{E}\left[|W|^2\right] - (\mathbb{E}\left[W\right])^2,$$

which reduces, in the case of $W$ symmetrically distributed around 0, to $\mathbb{E}\left[|W|^2\right]$. We do not know yet the value of $\text{Var}(W) = \mathbb{E}\left[|W|^2\right]$. However, we do know a related quantity, $\text{Var}(|W|)$, because the magnitude of complex normal values, $|W|$, follows the Rayleigh distribution (Chi-distributed with two degrees of freedom (DOFs)). This quantity is

$$\text{Var}(|W|) = \mathbb{E}\left[|W||W|^*\right] - (\mathbb{E}\left[|W|\right])^2 = \mathbb{E}\left[|W|^2\right] - (\mathbb{E}\left[|W|\right])^2. \tag{9}$$

Putting them together:
$$\text{Var}(|W|) = \text{Var}(W) - (\mathbb{E}[|W|])^2, \quad \text{and} \quad \text{Var}(W) = \text{Var}(|W|) + (\mathbb{E}[|W|])^2.$$

We now have a formulation for the variance of $W$ in terms of the variance and expectation of its magnitude, both properties analytically computable from the Rayleigh distribution's single parameter, $\sigma$, indicating the *mode*. These are:

$$\mathbb{E}[|W|] = \sigma\sqrt{\frac{\pi}{2}}, \quad \text{Var}(|W|) = \frac{4-\pi}{2}\sigma^2.$$

The variance of $W$ can thus be expressed in terms of its generating Rayleigh distribution's single parameter, $\sigma$, thus:

$$\text{Var}(W) = \frac{4-\pi}{2}\sigma^2 + \left(\sigma\sqrt{\frac{\pi}{2}}\right)^2 = 2\sigma^2. \tag{10}$$

If we want to respect the Glorot and Bengio (2010) criterion which ensures that the variances of the input, the output and their gradients are the same, then we would have $\text{Var}(W) = 2/(n_{in} + n_{out})$, where $n_{in}$ and $n_{out}$ are the number of input and output units respectively. In such case, $\sigma = 1/\sqrt{n_{in} + n_{out}}$. If we want to respect the He et al. (2015b) initialization that presents an initialization criterion that is specific to ReLUs, then $\text{Var}(W) = 2/n_{in}$ which $\sigma = 1/\sqrt{n_{in}}$.

The magnitude of the complex parameter $W$ is then initialized using the Rayleigh distribution with the appropriate mode $\sigma$. We can see from equation 10, that the variance of $W$ depends on on its magnitude and not on its phase. We then initialize the phase using the uniform distribution between $-\pi$ and $\pi$. By performing the multiplication of the magnitude by the phasor as is detailed in equation 8, we perform the complete initialization of the complex parameter.

In all the experiments that we report, we use variant of this initialization which leverages the independence property of unitary matrices. As it is stated in Cogswell et al. (2015), Srivastava et al. (2014), and Tompson et al. (2015), learning decorrelated features is beneficial for learning as it allows to perform better generalization and faster learning. This motivates us to achieve initialization by considering a (semi-)unitary matrix which is reshaped to the size of the weight tensor. Once this is done, the weight tensor is mutiplied by $\sqrt{He_{var}/\text{Var}(W)}$ or $\sqrt{Glorot_{var}/\text{Var}(W)}$ where $Glorot_{var}$ and $He_{var}$ are respectively equal to $2/(n_{in} + n_{out})$ and $2/n_{in}$. In such a way we allow kernels to be independent from each other as much as possible while respecting the desired criterion. Note that we perform the analogous initialization for real-valued models by leveraging the independence property of orthogonal matrices in order to build kernels that are as much independent from each other as possible while respecting a given criterion.

## 3.7 COMPLEX CONVOLUTIONAL RESIDUAL NETWORK

A deep convolutional residual network of the nature presented in He et al. (2015a; 2016) consists of 3 *stages* within which feature maps maintain the same shape. At the end of a stage, the feature maps are downsampled by a factor of 2 and the number of convolution filters are doubled. The sizes of the convolution kernels are always set to 3 x 3. Within a stage, there are several *residual blocks* which comprise 2 convolution layers each. The contents of one such residual block in the real and complex setting is illustrated in Appendix Figure 1b.

In the complex valued setting, the majority of the architecture remains identical to the one presented in He et al. (2016) with a few subtle differences. Since all datasets that we work with have real-valued inputs, we present a way to learn their imaginary components to let the rest of the network operate in the complex plane. We learn the initial imaginary component of our input by performing the operations present within a single real-valued residual block

$$BN \rightarrow ReLU \rightarrow Conv \rightarrow BN \rightarrow ReLU \rightarrow Conv$$

Using this learning block yielded better emprical results than assuming that the input image has a null imaginary part. The parameters of this real-valued residual block are trained by backpropagating errors from the task specific loss function. Secondly, we perform a $Conv \rightarrow BN \rightarrow Activation$ operation on the obtained complex input before feeding it to the first residual block. We also perform the same operation on the real-valued network input instead of $Conv \rightarrow Maxpooling$ as in He et al. (2016). Inside, residual blocks, we subtly alter the way in which we perform a projection at

Table 1: Classification error on CIFAR-10, CIFAR-100 and SVHN* using different complex activations functions ($z$ReLU, modReLU and $\mathbb{C}$ReLU). WS, DN and IB stand for the wide and shallow, deep and narrow and in-between models respectively. The prefixes R & C refer to the real and complex valued networks respectively. Performance differences between the real network and the complex network using $\mathbb{C}$ReLU are reported between their respective best models. All models are constructed to have roughly 1.7M parameters except the modReLU models which have roughly 2.5M parameters. modReLU and $z$ReLU were largely outperformed by $\mathbb{C}$ReLU in the reported experiments. Due to limited resources, we haven't performed all possible experiments as the conducted ones are already conclusive. A "-" is filled in front of an unperformed experiment.

| ARCH | CIFAR-10 | | | CIFAR-100 | | | SVHN* | | |
|---|---|---|---|---|---|---|---|---|---|
| | $z$ReLU | modReLU | $\mathbb{C}$ReLU | $z$ReLU | modReLU | $\mathbb{C}$ReLU | $z$ReLU | modReLU | $\mathbb{C}$ReLU |
| CWS | 11.71 | 23.42 | 6.17 | - | 50.38 | **26.36** | 80.41 | 7.43 | 3.70 |
| CDN | 9.50 | 22.49 | 6.73 | - | 50.64 | 28.22 | 80.41 | - | 3.72 |
| CIB | 11.36 | 23.63 | 5.59 | - | 48.10 | 28.64 | 4.98 | - | 3.62 |
| | ReLU | | | ReLU | | | ReLU | | |
| RWS | **5.42** | | | 27.22 | | | **3.42** | | |
| RDN | 6.29 | | | 27.84 | | | 3.52 | | |
| RIB | 6.07 | | | 27.71 | | | 4.30 | | |
| DIFF | -0.17 | | | +0.86 | | | -0.20 | | |

Table 2: Classification error on CIFAR-10, CIFAR-100 and SVHN* using different normalization strategies. NCBN, CBN and BN stand for a Naive variant of the complex batch-normalization, complex batch-normalization and regular batch normalization respectively. (R) & (C) refer to the use of the real- and complex-valued convolution respectively. The complex models use $\mathbb{C}$ReLU as activation. All models are constructed to have roughly 1.7M parameters. 5 out of 6 experiments using the naive variant of the complex batch normalization failed with the apparition of NaNs during training. As these experiments are already conclusive and due to limited resources, we haven't conducted other experiments for the NCBN model. A "-" is filled in front of an unperformed experiment.

| ARCH | CIFAR-10 | | | CIFAR-100 | | | SVHN* | | |
|---|---|---|---|---|---|---|---|---|---|
| | NCBN(C) | CBN(R) | BN(C) | NCBN(C) | CBN(R) | BN(C) | NCBN(C) | CBN(R) | BN(C) |
| WS | - | **5.47** | 6.32 | 27.29 | **26.63** | 27.89 | NAN | 3.80 | **3.52** |
| DN | - | 5.89 | 6.71 | NAN | 27.13 | 28.83 | NAN | 3.54 | 3.58 |
| IB | - | 5.66 | 6.83 | NAN | 26.99 | 29.89 | NAN | 3.74 | 3.56 |

the end of a stage in our network. We concatenate the output of the last residual block with the output of a 1x1 convolution applied on it with the same number of filters used throughout the stage and subsample by a factor of 2. In contrast, He et al. (2016) perform a similar 1x1 convolution with twice the number of feature filters in the current stage to both downsample the feature maps spatially and double them in number.

## 4 EXPERIMENTAL RESULTS

In this section, we present empirical results from using our model to perform image, music classification and spectrum prediction. First, we present our model's architecture followed by the results we obtained on CIFAR-10, CIFAR-100, and SVHN* as well as the results on automatic music transcription on the MusicNet benchmark and speech spectrum prediction on TIMIT.

### 4.1 IMAGE RECOGNITION

We adopt an architecture inspired by He et al. (2016). The latter will also serve as a baseline to compare against. We train comparable real-valued Neural Networks using the standard ReLU

activation function. We have tested our complex models with the $\mathbb{C}$ReLU, $z$ReLU and modRelu activation functions. We use a cross entropy loss for both real and complex models. A global average pooling layer followed by a single fully connected layer with a softmax function is used to classify the input as belonging to one of 10 classes in the CIFAR-10 and SVHN datasets and 100 classes for CIFAR-100.

We consider architectures that trade-off model depth (number of residual blocks per stage) and width (number of convolutional filters in each layer) given a fixed parameter budget. Specifically, we build three different models - wide and shallow (WS), deep and narrow (DN) and in-between (IB). In a model that has roughly 1.7 million parameters, our WS architecture for a complex network starts with 12 complex filters (24 real filters) per convolution layer in the initial stage and 16 residual blocks per stage. The DN architecture starts with 10 complex filters and 23 blocks per stage while the IB variant starts with 11 complex filters and 19 blocks per stage. The real-valued counterpart has also 1.7 million parameters. Its WS architecture starts with 18 real filters per convolutional layer and 14 blocks per stage. The DN architecture starts with 14 real filters and 23 blocks per stage and the IB architecture starts with 16 real filters and 18 blocks per stage.

All models (real and complex) were trained using the backpropagation algorithm with Stochastic Gradient Descent with Nesterov momentum (Nesterov, 1983) set at 0.9. We also clip the norm of our gradients to 1. We tweaked the learning rate schedule used in He et al. (2016) in both the real and complex residual networks to extract small performance improvements in both. We start our learning rate at 0.01 for the first 10 epochs to warm up the training and then set it at 0.1 from epoch 10-100 and then anneal the learning rates by a factor of 10 at epochs 120 and 150. We end the training at epoch 200.

Table 6.4 presents our results on performing image classification on CIFAR-10, CIFAR-100. In addition, we also consider a truncated version of the Street View House Numbers (SVHN) dataset which we call SVHN[*]. For computational reasons, we use the required 73,257 training images of Street View House Numbers (SVHN). We still test on all 26,032 images. For all the tasks and for both the real- and complex-valued models, The WS architecture has yielded the best performances. This is in concordance with Zagoruyko and Komodakis (2016) who observed that wider and shallower residual networks perform better than their deeper and narrower counterpart. On CIFAR-10 and SVHN[*], the real-valued representation performs slightly better than its complex counterpart. On CIFAR-100, the complex representation outperforms the real one. In general, the obtained results for both representation are quite comparable. To understand the effect of using either real or complex representation for a given task, we designed hybrid models that combine both. Table 2 contains the results for hybrid models. We can observe in the Table 2 that in cases where complex representation outperformed the real one (wide and shallow on CIFAR-100), combining a real-valued convolutional filter with a complex batch normalization improves the accuracy of the real-valued convolutional model. However, the complex-valued one is still outperforming it. In cases, where real-valued representation outperformed the complex one (wide and shallow on CIFAR-10 and SVHN[*]), replacing a complex batch normalization by a regular one increased the accuracy of the complex convolutional model. Despite that replacement, the real-valued model performs better in terms of accuracy for such tasks. In general, these experiments show that the difference in efficiency between the real and complex models varies according to the dataset, to the task and to the architecture.

Ablation studies were performed in order to investigate the importance of the 2D whitening operation that occurs in the complex batch normalization. We replaced the complex batch normalization layers with a naive variant (NCBN) which, instead of left multiplying the centred unit by the inverse square root of its covariance matrix, just divides it by its complex variance. Here, this naive variant of CBN is Mimicking the regular BN by not taking into account correlation between the elements in the complex unit. The Naive variant of the Complex Batch Normalization performed very poorly; In 5 out of 6 experiments, training failed with the appearance of NaNs (See Section 6.6 for the explanation). By way of contrast, all 6 complex-valued Batch Normalization experiments converged. Results are given in Table 2.

Another ablation study was undertaken to compare $\mathbb{C}$ReLU, modReLU and $z$RELU. Again the differences were stark: All $\mathbb{C}$ReLU experiments converged and outperformed both modReLU and $z$ReLU, both which variously failed to converge or fared substantially worse. We think that modRelu didn't perform as well as $\mathbb{C}$ReLU due to the fact that consecutive layers in a feed-forward net do not represent time-sequential patterns, and so, they might need to drop some phase information. Results

Table 3: MusicNet experiments. *FS* is the sampling rate. *Params* is the total number of parameters. We report the average precision (AP) metric that is the area under the precision-recall curve.

| ARCHITECTURE | FS | PARAMS | AP, % |
|---|---|---|---|
| SHALLOW, REAL | 11KHZ | | 66.1 |
| SHALLOW, COMPLEX | 11KHZ | | 66.0 |
| SHALLOW, THICKSTUN ET AL. (2016) | 44.1KHZ | - | 67.8 |
| DEEP, REAL | 11KHZ | 10.0M | 69.6 |
| DEEP, COMPLEX | 11KHZ | 8.8M | **72.9** |

are reported in Table 6.4. More discussion about phase information encoding is presented in section 6.7.

## 4.2 AUTOMATIC MUSIC TRANSCRIPTION

In this section we present results for the automatic music transcription (AMT) task. The nature of an audio signal allows one to exploit complex operations as presented earlier in the paper. The experiments were performed on the MusicNet dataset (Thickstun et al., 2016). For computational efficiency we resampled the original input from 44.1kHz to 11kHz using the algorithm described in Smith (2002). This sampling rate is sufficient to recognize frequencies presented in the dataset while reducing computational cost dramatically. We modeled each of the 84 notes that are present in the dataset with independent sigmoids (due to the fact that notes can fire simultaneously). We initialized the bias of the last layer to the value of -5 to reflect the distribution of silent/non-silent notes. As in the baseline, we performed experiments on the raw signal and the frequency spectrum. For complex experiments with the raw signal, we considered its imaginary part equal to zero. When using the spectrum input we used its complex representation (instead of only the magnitudes, as usual for AMT) for both real and complex models. For the real model, we considered the real and imaginary components of the spectrum as separate channels. The model we used for raw signals is a shallow convolutional network similar to the model used in the baseline, with the size reduced by a factor of 4 (corresponding to the reduction of the sampling rate). The filter size was 512 samples (about 12ms) with a stride of 16. The model for the spectral input is similar to the VGG model (Simonyan and Zisserman, 2015). The first layer has filter with size of 7 and is followed by 5 convolutional layers with filters of size 3. The final convolution block is followed by a fully connected layer with 2048 units. The latter is followed, in its turn, by another fully connected layer with 84 sigmoidal units. In all of our experiments we use an input window of 4096 samples or its corresponding FFT (which corresponds to the 16,384 window used in the baseline) and predicted notes in the center of the window. All networks were optimized with Adam. We start our learning rate at $10^{-3}$ for the first 10 epochs and then anneal it by a factor of 10 at each of the epochs 100, 120 and 150. We end the training at epoch 200. For the real-valued models, we have used ReLU as activation. $\mathbb{C}$ReLU has been used as activation for the complex-valued models.

The complex network was initialized using the unitary initialization scheme respecting the He criterion as described in Section 3.6. For the real-valued network, we have used the analogue initialization of the weight tensor. It consists of performing an orthogonal initialization with a gain of $\sqrt{2}$. The complex batch normalization was applied according to Section 3.5. Following Thickstun et al. (2016) we used recordings with ids '2303', '2382', '1819' as the test subset and additionally we created a validation subset using recording ids '2131', '2384', '1792', '2514', '2567', '1876' (randomly chosen from the training set). The validation subset was used for model selection and early stopping. The remaining 321 files were used for training. The results are summarized on Table 3. We achieve a performance comparable to the baseline with the shallow convolutional network. our VGG-based deep real-valued model reaches $69.6\%$ average precision on the downsampled data. With significantly fewer parameters than its real counterpart, the VGG-based deep complex model, achieves $72.9\%$ average precision which is the state of the art to the best of our knowledge. See Figures 2 and 3 in the Appendix for precision-recall curves and a sample of the output of the model.

Table 4: Speech Spectrum Prediction on TIMIT test set. *CConv-LSTM* denotes the Complex Convolutional LSTM.

| MODEL | #PARAMS | MSE(VALIDATION) | MSE(TEST) |
|---|---|---|---|
| LSTM WISDOM ET AL. (2016) | $\approx 135$K | 16.59 | 16.98 |
| FULL-CAPACITY URNN WISDOM ET AL. (2016) | $\approx 135$K | 14.56 | 14.66 |
| CONV-LSTM (OUR BASELINE) | $\approx 88$K | 11.10 | 12.18 |
| CCONV-LSTM (OURS) | $\approx 88$K | **10.78** | **11.90** |

### 4.3 SPEECH SPECTRUM PREDICTION

We apply both a real Convolutional LSTM Xingjian et al. (2015) and a complex Convolutional LSTM on speech spectrum prediction task (See section 6.5 in the Appendix for the details of the real and complex Convolutional LSTMs). In this task, the model predicts the magnitude spectrum. It implicitly infers the real and imaginary components of the spectrum at time $t + 1$, given all the spectrum (imaginary part and real components) up to time $t$. This is slightly different from (Wisdom et al., 2016). The real and imaginary components are considered as separate channels in both model. We evaluate the model with mean-square-error (MSE) on log-magnitude to compare with the others Wisdom et al. (2016). The experiments are conducted on a downsampled (8kHz) version of the TIMIT dataset. By following the steps in Wisdom et al. (2016), raw audio waves are transformed into frequency domain via short-time Fourier transform (STFT) with a Hann analysis window of 256 samples and a window hop of 128 samples (50% overlap). We use a training set with 3690 utterances, a validation set with 400 utterances and a standard test set with 192 utterance.

To match the number of parameters for both model, the Convolutional LSTM has 84 feature maps while the complex model has 60 complex feature maps (120 feature maps in total). Adam Kingma and Ba (2014) with a fixed learning rate of 1e-4 is used in both experiments. We initialize the complex model with the unitary initialization scheme and the real model with orthogonal initialization respecting the Glorot criterion. The result is shown in Table 4 and the learning curve is shown in Figure 4. Our baseline model has achieved the state of the art and the complex convolutional LSTM model performs better over the baseline in terms of MSE and convergence.

## 5 CONCLUSIONS

We have presented key building blocks required to train complex valued neural networks, such as complex batch normalization and complex weight initialization. We have also explored a wide variety of complex convolutional network architectures, including some yielding competitive results for image classification and state of the art results for a music transcription task and speech spectrum prediction. We hope that our work will stimulate further investigation of complex valued networks for deep learning models and their application to more challenging tasks such as generative models for audio and images.

### ACKNOWLEDGEMENTS

We are grateful to Roderick Murray-Smith, Jörn-Henrik Jacobsen, Jesse Engel and all the students at MILA, especially Jason Jo, Anna Huang and Akram Erraqabi for helpful feedback and discussions. We also thank the developers of Theano (Theano Development Team, 2016) and Keras (Chollet et al., 2015). We are grateful to Samsung and the Fonds de Recherche du Québec – Nature et Technologie for their financial support. We would also like to acknowledge NVIDIA for donating a DGX-1 computer used in this work.

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

# 6  APPENDIX

In practice, the complex convolution operation is implemented as illustrated in Fig.1a where $M_I$, $M_R$ refer to imaginary and real feature maps and $K_I$ and $K_R$ refer to imaginary and real kernels. $M_I K_I$ refers to result of a real-valued convolution between the imaginary kernels $K_I$ and the imaginary feature maps $M_I$.

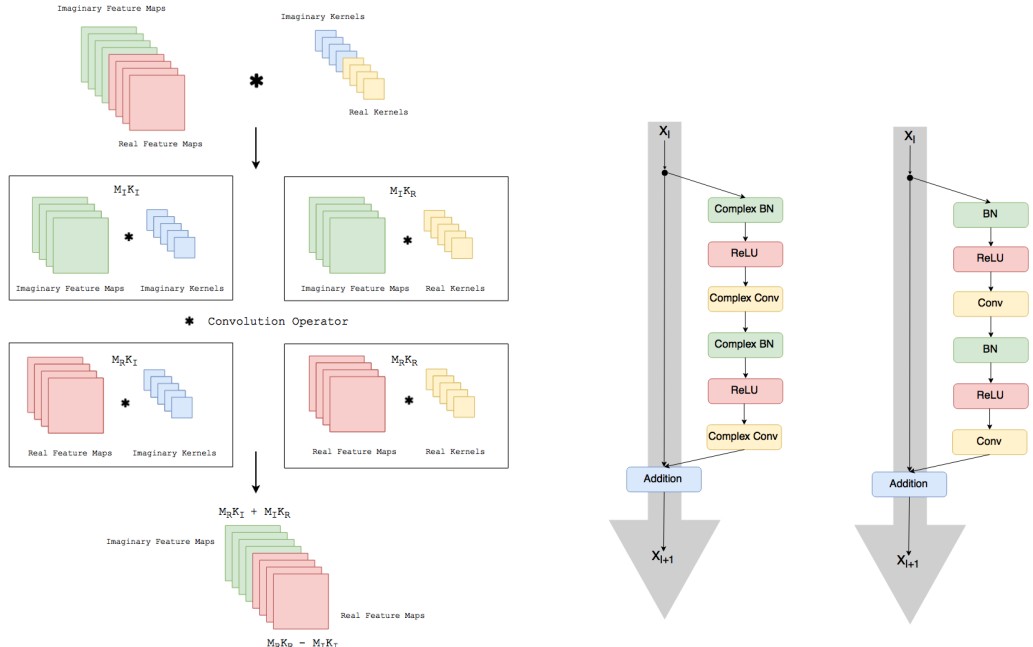

(a) An illustration of the complex convolution operator.

(b) A complex convolutional residual network (left) and an equivalent real-valued residual network (right).

Figure 1: Complex convolution and residual network implementation details.

## 6.1  MUSICNET ILLUSTRATIONS

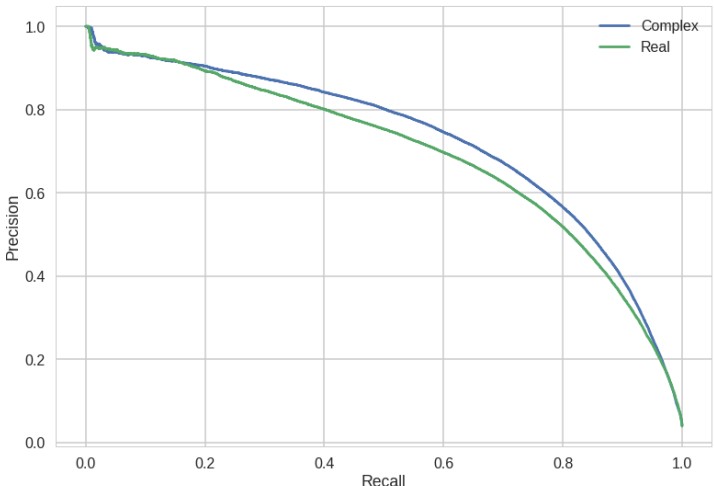

Figure 2: Precision-recall curve

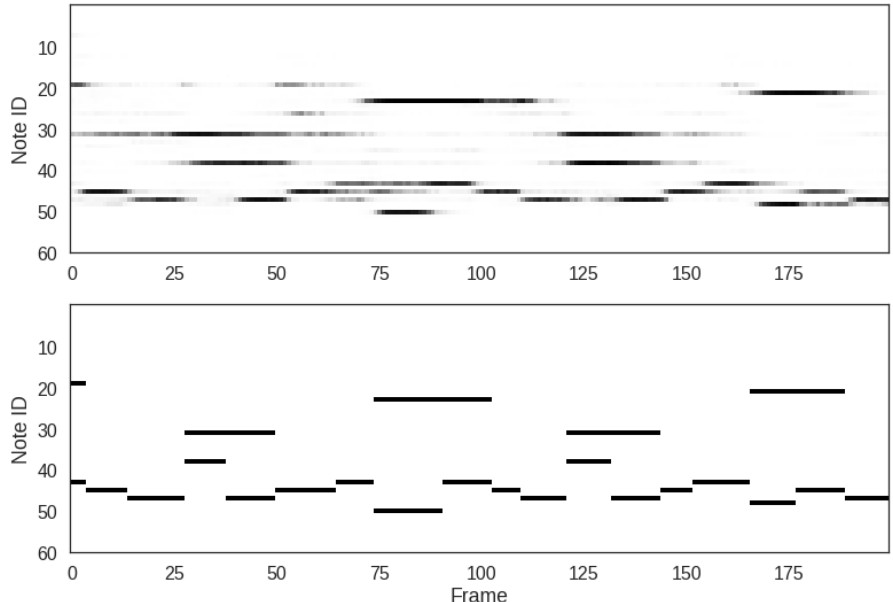

Figure 3: Predictions (Top) vs. ground truth (Bottom) for a music segment from the test set.

## 6.2 HOLOMORPHISM AND CAUCHY–RIEMANN EQUATIONS

*Holomorphism*, also called *analyticity*, ensures that a complex-valued function is *complex differentiable* in the neighborhood of every point in its domain. This means that the derivative, $f'(z_0) \equiv \lim_{\Delta z \to 0} \left[ \frac{(f(z_0) + \Delta z) - f(z_0)}{\Delta z} \right]$ of $f$, exists at every point $z_0$ in the domain of $f$ where $f$ is a complex-valued function of a complex variable $z = x + i y$ such that $f(z) = u(x, y) + i v(x, y)$. $u$ and $v$ are real-valued functions. One possible way of expressing $\Delta z$ is to have $\Delta z = \Delta x + i \Delta y$. $\Delta z$ can approach 0 from multiple directions (along the real axis, imaginary axis or in-between). However, in order to be complex differentiable, $f'(z_0)$ must be the same complex quantity regardless of direction of approach. When $\Delta z$ approaches 0 along the real axis, $f'(z_0)$ could be written as:

$$
\begin{aligned}
f'(z_0) &\equiv \lim_{\Delta z \to 0} \left[ \frac{(f(z_0) + \Delta z) - f(z_0)}{\Delta z} \right] \\
&= \lim_{\Delta x \to 0} \lim_{\Delta y \to 0} \left[ \frac{\Delta u(x_0, y_0) + i \, \Delta v(x_0, y_0)}{\Delta x + i \, \Delta y} \right] \\
&= \lim_{\Delta x \to 0} \left[ \frac{\Delta u(x_0, y_0) + i \, \Delta v(x_0, y_0)}{\Delta x + i \, 0} \right].
\end{aligned}
\tag{11}
$$

When $\Delta z$ approaches 0 along the imaginary axis, $f'(z_0)$ could be written as:

$$
\begin{aligned}
&= \lim_{\Delta y \to 0} \lim_{\Delta x \to 0} \left[ \frac{\Delta u(x_0, y_0) + i \, \Delta v(x_0, y_0)}{\Delta x + i \, \Delta y} \right] \\
&= \lim_{\Delta y \to 0} \left[ \frac{\Delta u(x_0, y_0) + i \, \Delta v(x_0, y_0)}{0 + i \, \Delta y} \right]
\end{aligned}
\tag{12}
$$

Satisfying equations 11 and 12 is equivalent of having $\frac{\partial f}{\partial z} = \frac{\partial u}{\partial x} + i \frac{\partial v}{\partial x} = -i \frac{\partial u}{\partial y} + \frac{\partial v}{\partial y}$. So, in order to be complex differentiable, $f$ should satisfy $\frac{\partial u}{\partial x} = \frac{\partial v}{\partial y}$ and $\frac{\partial u}{\partial y} = -\frac{\partial v}{\partial x}$. These are called the Cauchy–Riemann equations and they give a necessary condition for $f$ to be complex differentiable or "*holomorphic*". Given that $u$ and $v$ have continuous first partial derivatives, the Cauchy-Riemann equations become a sufficient condition for $f$ to be holomorphic.

### 6.3 The Genralized Complex Chain Rule for a Real-Valued Loss Function

If $L$ is a real-valued loss function and $z$ is a complex variable such that $z = x + i\,y$ where $x, y \in \mathbb{R}$, then:

$$\nabla_L(z) = \frac{\partial L}{\partial z} = \frac{\partial L}{\partial x} + i\frac{\partial L}{\partial y} = \frac{\partial L}{\partial \Re(z)} + i\frac{\partial L}{\partial \Im(z)} = \Re(\nabla_L(z)) + i\Im(\nabla_L(z)). \qquad (13)$$

Now if we have another complex variable $t = r + i\,s$ where $z$ could be expressed in terms of $t$ and $r, s \in \mathbb{R}$, we would then have:

$$\begin{aligned}
\nabla_L(t) &= \frac{\partial L}{\partial t} = \frac{\partial L}{\partial r} + i\frac{\partial L}{\partial s} \\
&= \frac{\partial L}{\partial x}\frac{\partial x}{\partial r} + \frac{\partial L}{\partial y}\frac{\partial y}{\partial r} + i\left(\frac{\partial L}{\partial x}\frac{\partial x}{\partial s} + \frac{\partial L}{\partial y}\frac{\partial y}{\partial s}\right) \\
&= \frac{\partial L}{\partial x}\left(\frac{\partial x}{\partial r} + i\frac{\partial x}{\partial s}\right) + \frac{\partial L}{\partial y}\left(\frac{\partial y}{\partial r} + i\frac{\partial y}{\partial s}\right) \\
&= \frac{\partial L}{\partial \Re(z)}\left(\frac{\partial x}{\partial r} + i\frac{\partial x}{\partial s}\right) + \frac{\partial L}{\partial \Im(z)}\left(\frac{\partial y}{\partial r} + i\frac{\partial y}{\partial s}\right) \\
&= \Re(\nabla_L(z))\left(\frac{\partial x}{\partial r} + i\frac{\partial x}{\partial s}\right) + \Im(\nabla_L(z))\left(\frac{\partial y}{\partial r} + i\frac{\partial y}{\partial s}\right).
\end{aligned} \qquad (14)$$

### 6.4 Computational Complexity and FLOPS

In terms of computational complexity, the convolutional operation and the complex batchnorm are of the same order as their real counterparts. However, as a complex multiplication is 4 times more expensive than its real counterpart, all complex convolutions are 4 times more expensive as well.

Additionally, the complex BatchNorm is not implemented in cuDNN and therefore had to be simulated with a sizeable sequence of elementwise operations. This leads to a ballooning of the number of nodes in the compute graph and to inefficiencies due to lack of effective operation fusion. A dedicated cuDNN kernel will, however, reduce the cost to little more than that of the real-valued BatchNorm.

Ignoring elementwise operations, which constitute a negligible fraction of the floating-point operations in the neural network, we find that for all architectures in and for all of CIFAR10, CIFAR100 or SVHN, the inference cost in real FLOPS per example is roughly identical. It is $\sim 265$ MFLOPS for the $\mathbb{R}$-valued variant and $\sim 1030$ MFLOPS for the $\mathbb{C}$-valued variant of the architecture, approximately quadruple.

### 6.5 Convolutional LSTM

A Convolutional LSTM is similar to a fully connected LSTM. The only difference is that, instead of using matrix multiplications to perform computation, we use convolutional operations. The computation in a real-valued Convolutional LSTM is defined as follows:

$$\begin{aligned}
\mathbf{i}_t &= \sigma(\mathbf{W}_{xi} * \mathbf{x}_t + \mathbf{W}_{hi} * \mathbf{W}_{t-1} + \mathbf{b}_i) \\
\mathbf{f}_t &= \sigma(\mathbf{W}_{xf} * \mathbf{x}_t + \mathbf{W}_{hf} * \mathbf{h}_{t-1} + \mathbf{b}_f) \\
\mathbf{c}_t &= \mathbf{f}_t \circ \mathbf{c}_{t-1} + \mathbf{i}_t \circ \tanh(\mathbf{W}_{xc} * \mathbf{x}_t + \mathbf{W}_{hc} * \mathbf{h}_{t-1} + \mathbf{b}_c) \\
\mathbf{o}_t &= \sigma(\mathbf{W}_{xo} * \mathbf{x}_t + \mathbf{W}_{ho} * \mathbf{h}_{t-1} + \mathbf{b}_o) \\
\mathbf{h}_t &= \mathbf{o}_t \circ \tanh(\mathbf{c}_t)
\end{aligned} \qquad (15)$$

Where $\sigma$ denotes the sigmoidal activation function, $\circ$ the elementwise multiplication and $*$ the real-valued convolution. $\mathbf{i}_t, \mathbf{f}_t, \mathbf{o}_t$ represent the vector notation of the input, forget and output gates respectively. $\mathbf{c}_t$ and $\mathbf{h}_t$ represent the vector notation of the cell and hidden states respectively. the gates and states in a ConvLSTM are tensors whose last two dimensions are spatial dimensions. For each of the gates, $\mathbf{W}_{xgate}$ and $\mathbf{W}_{hgate}$ are respectively the input and hidden kernels.

For the Complex Convolutional LSTM, we just replace the real-valued convolutional operation by its complex counterpart. We maintain the real-valued elementwise multiplication. The sigmoid and tanh are both performed separately on the real and the imaginary parts.

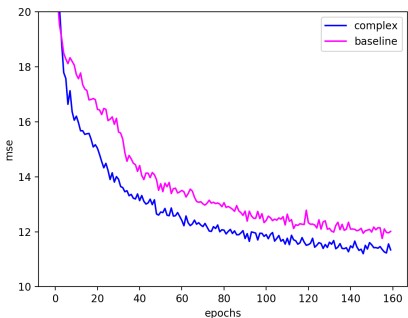

Figure 4: Learning curve for speech spectrum prediction from dev set.

## 6.6 COMPLEX STANDARDIZATION AND INTERNAL COVARIATE SHIFT

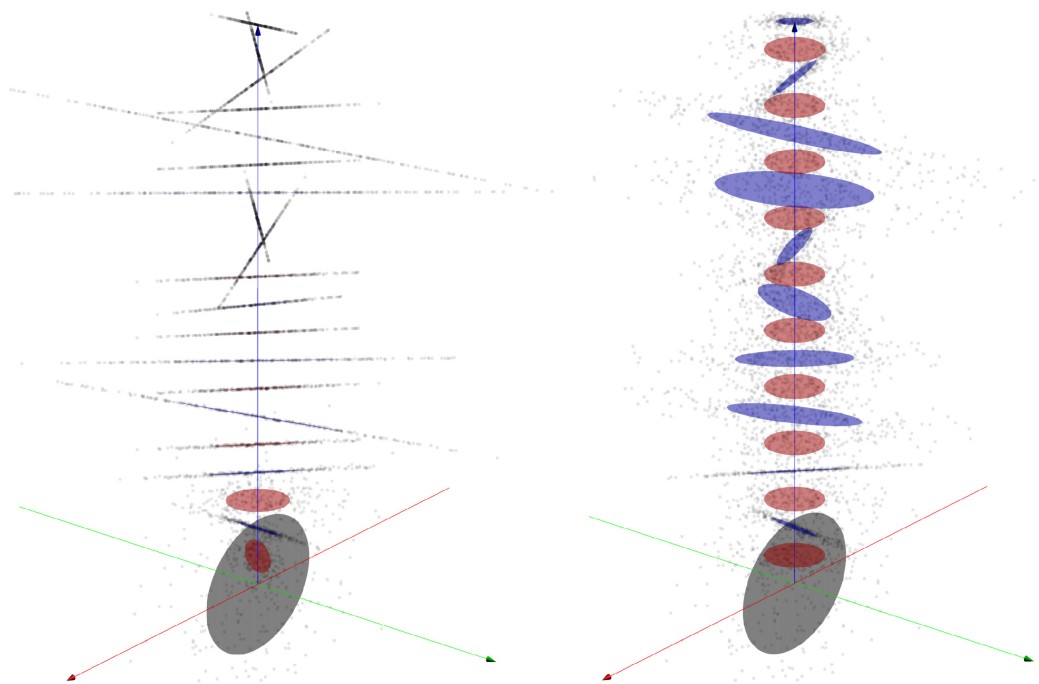

Figure 5: Depiction of Complex Standardization in Deep Complex Networks. *At left*, Naive Complex Standardization (division by complex standard deviation); *At right*, Complex Standardization (left-multiplication by inverse square root of covariance matrix between $\Re$ and $\Im$). The 250 input complex scalars are at the bottom, with $\Re(v)$ plotted on $x$ (red axis) and $\Im(v)$ plotted on $y$ (green axis). Deeper representations correspond to greater $z$ (blue axis). *The gray ellipse* encloses the input scalars within 1 standard deviation of the mean. *Red ellipses* enclose all scalars within 1 standard deviation of the mean after "standardization". *Blue ellipses* enclose all scalars within 1 standard deviation of the mean after left-multiplying all the scalars by a random $2 \times 2$ linear transformation matrix. With the naive standardization, the distribution becomes progressively more elliptical with every layer, eventually collapsing to a line. This ill-conditioning manifests itself as NaNs in the forward pass or backward pass. With the complex standardization, the points' distribution is always successfully re-circularized.

## 6.7 PHASE INFORMATION ENCODING

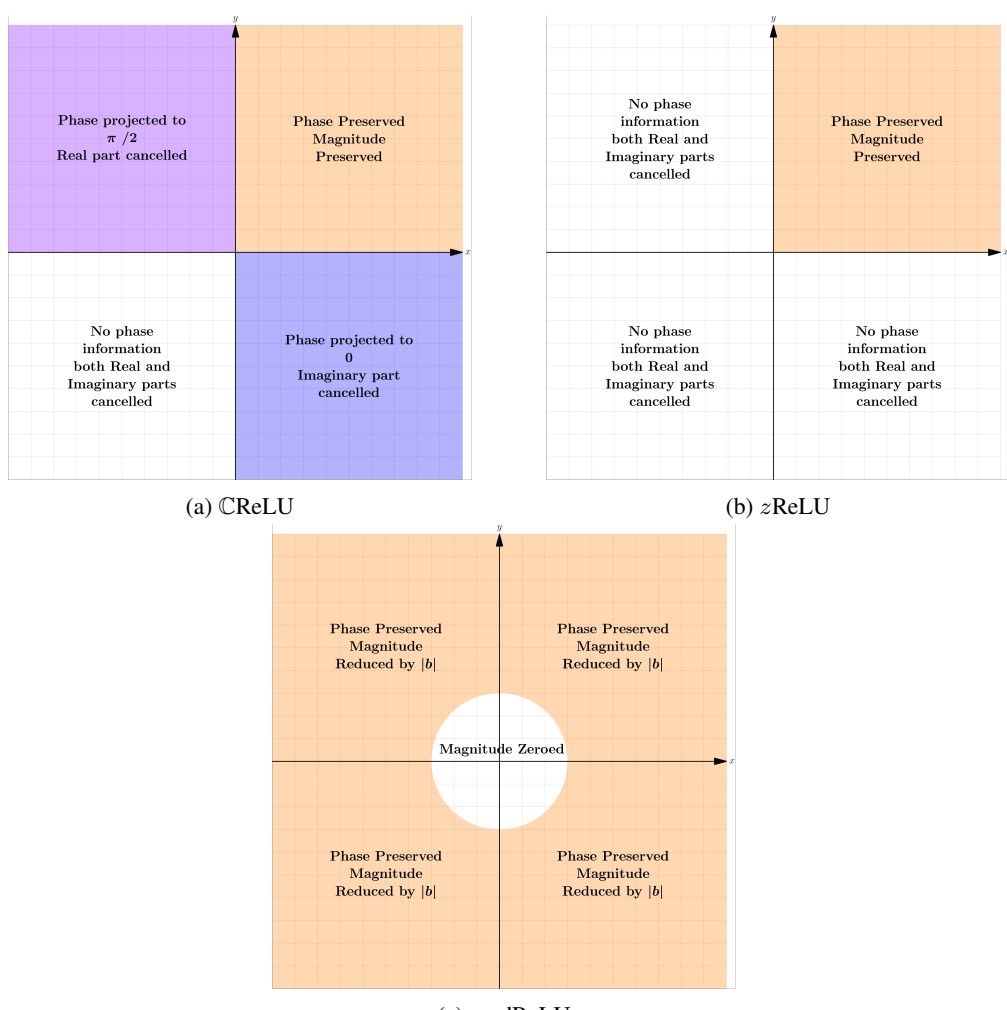

Figure 6: Phase information encoding for each of the activation functions tested for the Deep Complex Network. The x-axis represents the real part and the y-axis axis represents the imaginary part; The bottom figure corresponds to the case where $b < 0$ for modReLU. The radius of the white circle is equal to $|b|$. In case where $b \geq 0$, the whole complex plane would be preserving both phase and magnitude information and the whole plane would have been colored with orange. Different colors represents different encoding of the complex information in the plane. We can see the for both $z$ReLU and modReLU, the complex representation is discriminated into two regions, i.e, the one that preserves the whole complex information (colored in orange) and the one that cancels it (colored in white). However, $\mathbb{C}$ReLU discriminates the complex information into 4 regions where in two of which, phase information is projected and not canceled. This allows $\mathbb{C}$ReLU to discriminate information easier with respect to phase information than the other activation functions. For both $z$ReLU and modReLU, we can see that phase information may be preserved explicitly through a number of layers when these activation functions are operating in their linear regime, prior to a layer further up in a network where the phase of an input lies in a zero region. $\mathbb{C}$ReLU has more flexibility manipulating phase as it can either set it to zero or $\pi/2$, or even delete the phase information (when both real and imaginary parts are canceled) at a given level of depth in the network.

