# OpenReview forum: "Deep Complex Networks"
_ICLR.cc/2018/Conference — Accept (Poster)_

### Official Review · AnonReviewer1 · 2017-11-28
**Well-written, but unclear what happens with phase information**

**Rating:** 7
**Confidence:** 4

**Review:**

This paper defines building blocks for complex-valued convolutional neural networks: complex convolutions, complex batch normalisation, several variants of the ReLU nonlinearity for complex inputs, and an initialisation strategy. The writing is clear, concise and easy to follow.

An important argument in favour of using complex-valued networks is said to be the propagation of phase information. However, I feel that the observation that CReLU works best out of the 3 proposed alternatives contradicts this somewhat. CReLU simply applies ReLU component-wise to the real and imaginary parts, which has an effect on the phase information that is hard to conceptualise. It definitely does not preserve phase, like modReLU would.

This makes me wonder whether the "complex numbers" paradigm is applied meaningfully here, or whether this is just an arbitrary way of doing some parameter sharing in convnets that happens to work reasonably well (note that even completely random parameter tying can work well, as shown in "Compressing neural networks with the hashing trick" by Chen et al.). Some more insight into how phase information is used, what it represents and how it is propagated through the network would help to make sense of this.

The image recognition results are mostly inconclusive, which makes it hard to assess the benefit of this approach. The improved performance on the audio tasks seems significant, but how the complex nature of the networks helps achieve this is not really demonstrated. It is unclear how the phase information in the input waveform is transformed into the phase of the complex activations in the network (because I think it is implied that this is what happens). This connection is a bit vague. Once again, a more in-depth analysis of this phase behavior would be very welcome.

I'm on the fence about this work: I like the ideas and they are explained well, but I'm missing some insight into why and how all of this is actually helping to improve performance (especially w.r.t. how phase information is used).


Comments:

- The related work section is comprehensive but a bit unstructured, with each new paragraph seemingly describing a completely different type of work. Maybe some subsection titles would help make it feel a bit more cohesive.

- page 3: "(cite a couple of them)" should be replaced by some actual references :)

- Although care is taken to ensure that the complex and real-valued networks that are compared in the experiments have roughly the same number of parameters, doesn't the complex version always require more computation on account of there being more filters in each layer? It would be nice to discuss computational cost as well.


REVISION: I have decided to raise my rating from 5 to 7 as I feel that the authors have adequately addressed many of my comments. In particular, I really appreciated the additional appendix sections to clarify what actually happens as the phase information is propagated through the network.

Regarding the CIFAR results, I may have read over it, but I think it would be good to state even more clearly that these experiments constitute a sanity check, as both reviewer 1 and myself were seemingly unaware of this. With this in mind, it is of course completely fine that the results are not better than for real-valued networks.

---

> ### Public Comment · (anonymous) · 2018-01-05
> **Response to Reviewer 1**
>
> We thank the reviewer for the useful feedback. We have considered the comments and added a discussion on Phase Encoding in the appendix of the revised manuscript. We have illustrated the difference in encoding between the different activation functions tested for the deep complex network and shown that CReLU has more flexibility discriminating phase information. We also show that for all the tested activations, phase information is not necessarily preserved but, depending on where the complex representation lies in the complex plane, the latter might be either preserved, altered or discarded.
>
> “ CReLU simply applies ReLU component-wise to the real and imaginary parts, which has an effect on the phase information that is hard to conceptualise. It definitely does not preserve phase, like modReLU would.”
>
> “ … Some more insight into how phase information is used, what it represents and how it is propagated through the network would help to make sense of this.”
>
> Indeed, none of the ReLU based activations that are presented preserve phase completely. However, the nature of a ReLU is that it operates as an identity function in certain regions of the input. Therefore phase is preserved exactly in some regions of the input space and discriminated in others.
>
> For example, in section 3.4 we discuss the properties of the MoDReLU, CReLU and zReLU activation functions. We have added Section 6.6 which discusses the ways in which phase is preserved and manipulated in each of these cases.
>
> Importantly, in cases when activation functions do not preserve phase information, phase information can still influence subsequent computation. For example phase information may be preserved explicitly through a number of layers when activation functions are operating in their linear regime, prior to a layer further up in a network where the phase of an input lies in a zero region of an activation function.  In audio classification tasks, one can easily imagine how phase information could be important to use to influence classification decisions, but this does not mean that phase must be preserved all the way from input to the final classification output.
>
> “The image recognition results are mostly inconclusive, which makes it hard to assess the benefit of this approach.”
>
> As mentioned to reviewer 3. we feel it is important to underscore that our experiments on CIFAR with complex ResNets are included to demonstrate that our implementation is correct and that it yields results that are comparable to state of the art real architectures on a standard, well-known vision benchmark. This type of experiment is important because a naively implemented complex variation of a ResNet is *not* stable. Our complex batch norm formulation is essential to making deep complex networks work and therefore is an important contribution. (See Section 6.5).
>
> “The improved performance on the audio tasks seems significant, but how the complex nature of the networks helps achieve this is not really demonstrated. It is unclear how the phase information in the input waveform is transformed into the phase of the complex activations in the network (because I think it is implied that this is what happens). This connection is a bit vague. Once again, a more in-depth analysis of this phase behavior would be very welcome.”
>
> Consider also our experiments in Table 4 which use tanh and sigmoid activations. These activation functions are bijective (invertible) functions and therefore they allow phase information to pass through the network with negligible loss of information. Our experiments in Table 4 involve predicting spectrograms and therefore the utility of preserving phase information should be easy for the reader to imagine. Our experiments support this intuition, as we see clear performance improvements with these phase information preserving models compared to similar real computation only networks.
>
> “ Although care is taken to ensure that the complex and real-valued networks that are compared in the experiments have roughly the same number of parameters, doesn't the complex version always require more computation on account of there being more filters in each layer? It would be nice to discuss computational cost as well.”
>
> In terms of computational complexity, the convolutional operation and the complex batchnorm are of the same order than their real counterparts. However, as the complex convolution is 4 times more expensive than its real counterpart and as the complex batchnorm is not implemented in cudnn, this makes a difference in terms of running time.

---

### Official Review · AnonReviewer2 · 2017-11-28
**An extensive framework for complex-valued neural networks is presented.**

**Rating:** 8
**Confidence:** 4

**Review:**

The paper presents an extensive framework for complex-valued neural networks. Related literature suggests a variety of motivations for complex valued neural networks: biological evidence, richer representation capacity, easier optimization, faster learning, noise-robust memory retrieval mechanisms and more.

The contribution of the current work does not lie in presenting significantly superior results, compared to the traditional real-valued neural networks, but rather in developing an extensive framework for applying and conducting research with complex-valued neural networks. Indeed, the most standard work nowadays with real-valued neural networks depends on a variety of already well-established techniques for weight initialization, regularization, activation function, convolutions, etc. In this work, the complex equivalent of many of these basics tools are developed, such as a number of complex activation functions, complex batch normalization, complex convolution, discussion of complex differentiability, strategies for complex weight initialization, complex equivalent of a residual neural network.

Empirical results show that the new complex-flavored neural networks achieve generally comparable performance to their real-valued counterparts, on a variety of different tasks. Then again, the major contribution of this work is not advancing the state-of-the-art on many benchmark tasks, but constructing a solid framework that will enable stable and solid application and research of these well-motivated models.

---

> ### Public Comment · (anonymous) · 2018-01-05
> **Response to Reviewer 2**
>
> We appreciate your encouraging comments, we thank you for the review -- and for acknowledging the contribution we have made by creating “a solid framework that will enable stable and solid application” [of models based on deep complex networks].
>
> We want to inform the reviewer that we have added the following sections to the paper. We have added:
>
> Section 6.3 in order to detail the complex chain rule for the reader.
> Section 6.4 contains the details about the complex LSTM.
> Section 6.5 has been added in order to illustrate the utility of our complex batch normalization procedure.
> And we have also added a discussion about the phase information encoding for each of the activation functions tested in our work. You can find the latter in section 6.6.

---

### Official Review · AnonReviewer3 · 2017-11-30
**Using complex numbers for neural networks , but why?**

**Rating:** 4
**Confidence:** 4

**Review:**

Authors present complex valued analogues of real-valued convolution, ReLU and batch normalization functions. Their "related work section" brings up uses of complex valued computation such as discrete Fourier transforms and Holographic Reduced Representations. However their application don't seem to connect to any of those uses and simply reimplement existing real-valued networks as complex valued.

Their contributions are:

1. Formulate complex valued convolution
2. Formulate two complex-valued alternatives to ReLU and compare them
3. Formulate complex batch normalization as a "whitening" operation on complex domain
4. Formulate complex analogue of Glorot weight normalization scheme

Since any complex valued computation can be done with a real-valued arithmetic, switching to complex arithmetic needs a compelling use-case. For instance, some existing algorithm may be formulated in terms of complex values, and reformulating it in terms of real-valued computation may be awkward. However, cases the authors address, which are training batch-norm ReLU networks on standard datasets, are already formulated in terms of real valued arithmetic. Switching these networks to complex values doesn't seem to bring any benefit, either in simplicity, or in classification performance.

---

> ### Public Comment · (anonymous) · 2018-01-05
> **Response to Reviewer 3**
>
> We thank reviewer 3 for the useful feedback. We have clarified some important points about our work below -- in particular some of our experiments are included to demonstrate the correctness of our implementation on a well known benchmark and the importance of the building blocks for using complex numbers.
> We also feel it is important to draw special attention to the fact that we do indeed have clear quantitative improvements in performance using complex-valued neural networks compared to similar purely real-value networks on a well-defined audio task. This seems to have been overlooked in the initial review.
>
> “Their ‘related work section’ brings up uses of complex valued computation such as discrete Fourier transforms and Holographic Reduced Representations. However their application don't seem to connect to any of those uses”
>
> There is not a lot of work that has considered the use of complex representations in the context of deep learning. This is why we have cited and commented upon interesting examples of what people have used complex representations for in the past. Some examples of prior work include the work on holographic representations and spectral pooling which use the (complex) Fourier Transform. We therefore discuss them as related work, i.e, holographic representations (and Associative LSTMs) and Fourier Transform methods (with their applications to Spectral Representations for ConvNets).
>
> “Since any complex valued computation can be done with a real-valued arithmetic, switching to complex arithmetic needs a compelling use-case. For instance, some existing algorithm may be formulated in terms of complex values, and reformulating it in terms of real-valued computation may be awkward. However, cases the authors address, which are training batch-norm ReLU networks on standard datasets, are already formulated in terms of real valued arithmetic.”
>
> As mentioned to reviewer 1, we feel it is important to underscore that our experiments on CIFAR with complex ResNets are included to demonstrate that our implementation is correct and that it yields results that are comparable to state-of-the-art real-valued computation architectures on a standard, well-known vision benchmark. This type of experiment is important because a naively implemented complex variation of a ResNet is *not* stable. Our complex batch-norm formulation is essential to making deep complex networks work and therefore is an important contribution. (See Section 6.5)
>
> In light of Tables 3 and 4 we respectfully disagree with the following statement:
> “Switching these networks to complex values doesn't seem to bring any benefit, either in simplicity, or in classification performance.”
>
> Yes, for our ResNet experiments on CIFAR the complex network does not show a gain in performance; however, as we discuss above that was not the point of presenting those experiments. In contrast, our audio experiments in Tables 3 and 4 demonstrate the utility of complex VGG style architectures and complex convolutional LSTM architectures through clear performance gains. Comparing similarly structured real and complex networks, one sees increased performance for the complex variant in both of these sets of experiments. This is in line with the interpretation that for audio signals the use of complex neural networks allows information such as phase to be represented and manipulated within layers (implicitly in the case of rectangular complex numbers), and this yields higher performance compared to similarly structured real models.

---

> > ### Comment · AnonReviewer3 · 2018-01-12
> > **not clear there's a significant enough benefit to justify complexity**
> >
> > Unfortunately I'm not familiar with state of the art in music transcription.
> >
> > From description it sounds that test set is quite small (3 melodies). For a small test set, various hyper-parameters such as model architecture, learning rate schedule and choice of optimization algorithm are expected to have a strong impact. There's a number of hyper-parameters in the optimization, how were they chosen?
> >
> > It is not clear that the improvement is due to using complex numbers, rather than a particular choice of architecture/training procedure. This is the danger of using small/unpopular dataset -- improvement to state of the art may be due to chance or other uninteresting reasons.

---

### Public Comment · ~Jordan_Micah_Bennett1 · 2018-01-03
**Excellent Paper**

1. In Deep learning, it is typical for researchers to seek structures that can represent more information in weight space.

2. Real-valued convolutional neural nets can trivially store magnitude information.

3. Complex-valued convolutional neural nets can trivially store phase information, in addition to magnitude information. (As seen in the Deep Complex network paper)

4. So, seeking methods for representing more information in weight space, is a sensible, and typical goal in Deep learning, because the richer your weight space, the better the hypotheses or answers produced by neural net models!

---

### Decision · Program_Chairs · 2018-01-29
**ICLR 2018 Conference Acceptance Decision**

**Decision:**

Accept (Poster)

**Comment:**

The paper received mostly positive comments from experts. To summarize:

Pros:
-- The paper provides complex counterparts for typical architectures / optimization strategies used by real valued networks.
Cons:
-- Although the authors include plots explaining how nonlinearities transform phase, intuition about how phase gets processed can be improved.
-- Improving evaluations: Wisdom et al. computes log magnitude; real valued networks may not be suited for computing real / complex numbers which have a large dynamic range, like the complex spectra. So please compare performance by estimating magnitude as in Wisdom et al.
-- Please add computational cost, in terms of the number of multiplies and adds, to the final version of the paper.

I am recommending that the paper be accepted based on these reviews.